# An Update on Synthesis of Coumarin Sulfonamides as Enzyme Inhibitors and Anticancer Agents

**DOI:** 10.3390/molecules27051604

**Published:** 2022-02-28

**Authors:** Laila Rubab, Sumbal Afroz, Sajjad Ahmad, Saddam Hussain, Iram Nawaz, Ali Irfan, Fozia Batool, Katarzyna Kotwica-Mojzych, Mariusz Mojzych

**Affiliations:** 1Department of Chemistry, University of Lahore, Sargodha Campus, Sargodha 40100, Pakistan; lailarubabe95@gmail.com (L.R.); injoyia@gmail.com (I.N.); foziab1996@gmail.com (F.B.); 2Department of Chemistry, Government College University Lahore, Lahore 54000, Pakistan; afroz.sumbel@hotmail.com; 3Department of Chemistry, UET Lahore, Faisalabad Campus, Faisalabad 37630, Pakistan; sajjad.ahmad@uet.edu.pk; 4School of Biochemistry, Minhaj University Lahore, Lahore 54590, Pakistan; saddamhussainrao3@gmail.com; 5Abdul Razzaq Fazaia College, M.M. Alam Base Mianwali, Mainwali 42206, Pakistan; 6Department of Chemistry, Government College University Faisalabad, 5-Km, Jhang Road, Faisalabad 38040, Pakistan; 7Department of Histology, Embryology and Cytophysiology, Medical University of Lublin, Radziwiłłowska 11, 20-080 Lublin, Poland; katarzynakotwicamojzych@umlub.pl; 8Department of Chemistry, Siedlce University of Natural Sciences and Humanities, 3-go Maja 54, 08-110 Siedlce, Poland

**Keywords:** coumarin sulfonamide, synthesis, anticancer agents, carbonic anhydrase inhibitors, SAR

## Abstract

Coumarin is an important six-membered aromatic heterocyclic pharmacophore, widely distributed in natural products and synthetic molecules. The versatile and unique features of coumarin nucleus, in combination with privileged sulfonamide moiety, have enhanced the broad spectrum of biological activities. The research and development of coumarin, sulfonamide-based pharmacology, and medicinal chemistry have become active topics, and attracted the attention of medicinal chemists, pharmacists, and synthetic chemists. Coumarin sulfonamide compounds and analogs as clinical drugs have been used to cure various diseases with high therapeutic potency, which have shown their enormous development value. The diversified and wide array of biological activities such as anticancer, antibacterial, anti-fungal, antioxidant and anti-viral, etc. were displayed by diversified coumarin sulfonamides. The present systematic and comprehensive review in the current developments of synthesis and the medicinal chemistry of coumarin sulfonamide-based scaffolds give a whole range of therapeutics, especially in the field of oncology and carbonic anhydrase inhibitors. In the present review, various synthetic approaches, strategies, and methodologies involving effect of catalysts, the change of substrates, and the employment of various synthetic reaction conditions to obtain high yields is cited.

## 1. Introduction

Ever since the first time that coumarin **1** was isolated from natural source tonka bean (*Dipteryx odorata*), commonly known as cumaru, in 1820 by Vogel [1,2]. Coumarin is an old, important, and diversified oxygen containing six membered heterocyclic classes of 1,2 benzopyrones, which naturally occur in plants and many other species, such as fungi (*Armillariella tabescens*, *Fomitopsis officinalis*) and bacteria (*Streptomyces niveus*, *Escherichia coli*) [3,4]. More than 1300 coumarins are present in plants, which play vital role in physiology and overall functioning of plants [1]. The general structure of coumarin **1** is given below (Figure 1). In the mid-nineteenth century, the research and development of coumarin-based compounds and hybrid structures began via the famous Perkin condensation reaction between acetic anhydride and salicylaldehyde. The different synthetic classical techniques, such as Knoevenagel, Perkin and Pechmann reactions, are applied to achieve simple coumarins [5,6,7]. The rapid developments in the synthetic chemistry of coumarins have been made, due to their wide therapeutic potential as medicinal drugs. The coumarin scaffolds displayed an array of biological activities, such as coumarin chalcone derivatives, coumarin aryl sulfonamides, and coumarin hydrazine–hydrazone hybrids, etc., which were screened to investigate their anticancer activities [8,9,10,11,12]. Coumarin scaffolds are extensively studied for their antioxidant [13,14,15,16], antibacterial [17,18,19,20], anti-fungal [21,22,23], anti-inflammatory [24,25], anti-diabetic [26], vasorelaxant [27], analgesic [28], anti-HIV [29], antimicrobial [30], anti-coagulation [31], and anti-pyretic [32] activities, etc.

The combination of sulfonamide moieties with coumarin nucleus is attractive, as well as being a versatile platform for the research and development of a novel class of bioactive-targeted therapeutic agents [33]. The coumarin sulfonamides such as sulfocoumarin **2** and chlorophenyl-based coumarin sulfonamide **3** executed significant activity against carbonic anhydrase inhibitors IX and XII, while CAI17 coumarin sulfonamide scaffold **4** displayed remarkable anti-metastatic activity (Figure 2) [34,35,36].

This review article comprehensively reviews the overall current progress of coumarin sulfonamides in medicinal chemistry as carbonic anhydrase inhibitor and anticancer agents. The successful synthetic approaches, substitution pattern, and structure–activity relationship of different bioactive compounds are discussed.

## 2. Coumarin Sulfonamides as Anti-Cancer Agents and Carbonic Anhydrase Inhibitors

Cancer is one of the most lethal, notable complex and serious threats to human health, and has attracted attention worldwide. All over the world, about 7.6 million people die due to cancer every year, and around 13 million people will likely die before 2030. In 2020, globally, almost 10 million people died due to cancer [37,38]. Extensive research and development work have been conducted in the field of oncology to develop anticancer therapeutic agents, and large breakthroughs and great strides have been made over past 60 years [39]. Coumarin sulfonamide derivatives and analogs have therapeutic potential against different types of cancer cell lines and CAs (carbonic anhydrases). CAs are also known as carbonate dehydratases [34]. CAs are metalloenzymes which are present in all life forms, and are essential for equilibria between different simple but significant reaction species, such as carbon dioxide, proton, and bicarbonate [40,41,42,43,44]. In 1933, 88 years ago, these enzymes were discovered, and are still an extraordinary example of convergent evolution, and extensively studied and investigated for biomedical inhibitory activities. CAs were found in bacteria, archaea and eukarya; genetically, at least eight (α-, β-, γ-, δ-, ζ-, η-, θ- and ι-CAs) distinct families [40,41,42,43,44,45]. The α-CAs family is present in vertebrates, and the bacteria, algae, and cytoplasm of green plants, while β-CAs are found in bacteria, the chloroplasts of monodicotyledons and dicotyledons, and algae. The γ-CAs are mainly present in archaea and some bacteria, the δ-, ζ- and θ-CAs are present in some marine diatoms, and the η-CAs are present in protozoa. The ι-CAs were discovered in marine phytoplankton, as well as in some bacteria [45,46,47,48,49,50,51,52,53,54,55]. There are five membrane-bound isozymes: CA-IV, CA-IX, CA-XII, CA-XIV, and CA-XV), five cytosolic forms CA-I, CA-II, CA-III, CA-VII, and CA-XIII, a secreted CA isozyme CA-VI, and two mitochondrial forms, CA-VA and CA-VB [56,57,58,59,60]. CAs inhibition mechanism with coumarins was unraveled with kinetic and X-ray crystallographic techniques. The first natural product, coumarin, was bound to human isoform hCA-II, but the formation of the enzyme inhibitor complex is not a rapid process, it takes 6 h for incubation period, while other classes take just 15 min for the incubation period [61,62,63,64,65,66]. The coumarin sulfonamides’ anticancer and CAs inhibition activities are discussed below in more detail.

### 2.1. Benzenesulfonamide-Based Coumarins as Carbonic Anhydrases II and IX Inhibitors

Wang and coworkers designed a solvent-free green methodology to synthesize substituted coumarin containing sulfonamides derivatives, and screened for carbonic anhydrase inhibitory activities. In this synthetic strategy, Meldrum’s acid was reacted with various substituted phenol **5** to achieve substituted malonic acid-based mono phenol esters **6** in (91–94%) yield, which further cyclized with Eaton’s reagent under mild conditions to yield 4-hydroxycoumarin **7** in (79–91%) yield. In the next step, substituted 3-formyl-4-chlorocoumarin **8** (59–73%) was obtained by Vilsmeiere Haack reactions in dimethylformamide (DMF) and phosphoryl chloride. Derivative **8** was treated with substituted sulfonamides in ethanol at room temperature (rt) to 50 °C, leading to the formation of final coumarin sulfonamide derivative **9** in (45–79%) yield (Figure 1) [67].

The benzenesulfonamide coumarins’ eighteen derivatives were afforded and screened for their in vitro anticancer activity against mouse melanoma cells (B16–F10) and breast carcinoma cell lines (MCF-7), and two human carbonic anhydrase against hCAs II (cytosolic off target isoform) and hCAs IX (trans-membrane tumor-associated isoform). The IC_50_ calculations were done by using Origin 8.6 software using an inhibitory model with the sum of squares of the residuals minimized. In this study, the most active derivative was substituted dimethyl pyrimidine-based coumarin benzene sulfonamide **9a** (Figure 3), which displayed the highest and remarkable significant anticancer potential against MCF-7 cell lines with IC_50_ 0.0088 µM when compared with the reference drugs doxorubicin IC_50_ 0.072 µM and semaxanib IC_50_ 0.012 µM. Both the virtual screening and anticancer activity results for MCF-7 showed that the over-expressed CA might be the most active therapeutic candidate that coumarin sulfonamides interacted with. The substituted pyrimidine-based coumarin benzene sulfonamide **9b** (Figure 3) and di-*tert*-butyl substituted coumarin benzenesulfonamide containing pyrimidine **9c** (Figure 3) displayed strong inhibition against hCAs II and hCAs IX isoforms with IC_50_ values of 0.063 µM and 0.124 µM (Table 1) respectively, when compared with standard drugs acetazolamide (AAZ) and sulfanilamide (SA). The SAR studies investigated that the introduction of thiazole and methyl pyrimidine substitutions in the benzenesulfonyl ring of the coumarin enhanced the anticancer and carbonic anhydrase inhibition activities of the below-mentioned coumarin derivatives [67].

### 2.2. Thiazole-Sulfonamide Coumarin Hybrids as hCA I and hCA II Inhibitors

Kurt and colleagues developed a solvent-free approach to achieve the unsubstituted thiazole-based coumarin sulfonamides **17** by the reaction of 2-hydroxybenzaldehyde **10**, L-proline and ethyl 3-oxobutanoate **11**, by heating for 0.5 h at a temperature of 80–90 °C to obtain 3-acetylcoumarin **12** in 92% yield, which further refluxed for 15 min in chloroform and bromine solutions to obtain 3-(bromoacetyl) coumarin **13** in 98% yield. Refluxing compound **13** with thiourea **14** in ethanol for 1 h gives 2-amino coumarin thiazolyl derivatives **15** (90% yield) that were further treated with benzenesulfonyl chloride **16** derivatives at 60 °C in pyridine, which led to the synthesis of thiazole-based coumarin sulfonamides **17** in 68–82% yield (Figure 2) [68].

In this study, the thiazole ring of acetazolamide was combined with coumarin moiety to afford biologically active, substituted benzenesulfonamide-based coumaryl thiazole hybrids, and was screened for its anticancer activity against hCA I and hCA II (human carbonic anhydrase isoforms). Among all these compounds, the scaffold coumarin-thiazole-based naphthalene-2-sulpho-namide **17a** (Figure 4) displayed the strongest inhibition against hCA I and hCA II with the IC_50_ values 5.63 µM and 8.48 µM (Table 2), respectively. The SAR showed that bulky substituents such as s *tert*-butyl, naphthalene and iodine increase inhibitory activity, so compound **17a** showed the most potent inhibitory activity due to the steric effect of bulky group substitution, such as naphthalene on sulfonyl group against hCA I and hCA II [68].

### 2.3. Sulfonyl Ureido Coumarins Hybrids as Carbonic Anhydrase Inhibitors

Bozdag and collogues described a single step reaction to afford substituted sulfonyl ureido coumarins **20** in 53–88% yield by the treatment of coumarin **18** and sulfonyl ureido isocyanates **19** in acetonitrile (ACN) or dry acetone (Figure 3) [69].

The ary lsulfonylureido coumarin derivatives were evaluated for their inhibitory activity against hCA I and II (carbonic anhydrase cytosolic inhibitor) and hCA IX and XII (tumor-associated isoforms). The 4-chloro-substituted coumarin benzenesulfonamide **20a** (Figure 5) exhibited the highest inhibitory activity with a K_I_ value 20.2 nM against hCA IX and 6.0 nM against hCA XII (Table 3). Acetazolamide (AAZ) was used as a standard reference drug with K_I_ = 25.0 nM and K_I_ = 5.7 nM (Table 3) against hCA IX and hCA XII, respectively. The SAR showed that analogue **20a** was the most potent due to the presence of electron withdrawing Cl atom in the benzene ring of the sulfonyl ureido group [69].

### 2.4. Benzene Sulfonamido-Coumarinyl Hydrazones Hybrids as CA Inhibitors

Chandak et al., in 2016, synthesized sulfonamide bearing coumarin derivatives by a Hantzsch thiazole synthetic approach as shown in Figure 4. In this synthetic strategy, the thiazoyl hydrazine methylidene pyrazole **31** derivatives were achieved from 4-hydrazinobenzenesulfonamide hydrochloride, further converted into pyrazole-based carbaldehyde bearing thiosemicarbazones, and finally reacted to substituted bromoacetyl-based coumarins **30** by condensation reaction. In the second step, different 6-substituted 3-bromoacetylcoumarins **30** and 4-thioureido-benzenesulfonamide achieved 2-amino-substituted-coumarinylthiazoles **32** by condensation reaction. In the next step, heterocyclic series **33** containing three IBTs prepared by treatment of 2-aminobenzothiazole-6-sulfonamide that first obtained from sulfanilamide and 6-substituted-3-bromoacetylcoumarins. On the other hand, the derivatives of series 4, different 3-acetylcoumarins **29** and 4-hydrazinobenzenesulfonamide hydrochloride **34** by refluxing together in aqueous ethanol with anhydrous sodium acetate, give benzenesulfonamido-coumarinyl hydrazones, **35** (Figure 4) [70].

The following sulfonamide-bearing coumarin scaffold consisted of twenty-four compounds evaluated for the inhibition of hCA I, II, IX and XII (human carbonic anhydrase isoforms). Among all of these, the **32a** compound (Figure 6) exhibited strong potent inhibitory activity with a K_I_ value 2.28 nM (Table 4) against hCA IX, as compared to standard compound AZA with a K_I_ range 25.0 nM. Moreover, analogues **32a** and **32b** were most potent with K_I_ values 0.54 nM against hCA XII when compared to AZA with a K_I_ value 5.7 nM. The hybrid structure 4-{2-[1-(2-oxo-2*H*-chromen-3-yl)ethylidene]hydrazino} benzenesulfonamide **35a** revealed the highest activity K_I_ = 13.23 nM for hCA II in comparison with reference drug AZA with K_I_ value 12.1 nM. The compound 4-{2-[1-(6-bromo-2-oxo-2*H*-chromen-3-yl)ethylidene]hydrazino}benzenesulfonamide **35b** (Figure 6) screened potent inhibitory activity with a K_I_ value 21.95 nM against hCA I, as compared to standard compound AZA (acetazolamide) with a K_I_ range 250.0 nM (Table 4). The SAR showed that the introduction of bromo and unsubstituted H-atom on coumarin increase the carbonic anhydrase inhibitory activity of derivatives **35a** and **35b** (Figure 6), while the presence of electron-withdrawing Cl-atom and unsubstituted H-atom on coumarin enhances the inhibitory activity of compounds **32a** and **32b** [70].

### 2.5. Pyrazole-Based Coumarin Sulfonamides as CA Inhibitors and Anticancer Agents

Lu et al. in 2016 reported a synthetic approach in which acylhydrazone was converted to substituted pyrazole-based coumarin sulfonamide derivatives by consecutive reactions of various substrates (Figure 5). The substituted acetophenone **36** was treated with dimethyl oxalate using sodium methoxide in methanol for 6 h under refluxing conditions to achieve different substituted chalcones **37** on treatment with 4-hydrazinyl benzenesulfonamide, in the presence of methanol for 6 h, to afford scaffold **38**. In the second step, analogue **38** reacted with hydrazine monohydrate solution at 80 °C for 8 h to give substituted hydrazide derivatives **39**. The 4-chloro-coumarin-3-aldehyde **42** was afforded by treating 4-hydroxy coumarin **40** with a mixture of phosphoric trichloride and dimethyl formamide. In the last step, the derivatives **43** were obtained (60–90%) by the combination of derivative **39** and 4-chloro-coumarin-3-aldehyde **42** at room temperature for 12 h in ethanol and AcOH, as presented in Figure 5 [71].

The series of pyrazole moiety containing coumarin sulfonamides was tested for anti-proliferation activities in vitro, against four cancer cell lines (HeLa, HepG2, F10, A549) and two non-cancer cell lines (293T, L02). These synthetic scaffolds were also evaluated for inhibitory activities against various inhibitors, such as COX-2 and COX-1. The derivative **43a** (Figure 7) possessing most powerful anti-proliferative activity against HeLa, HepG2, F10, and A549 cell lines with IC_50_ values of 0.36 ± 0.05 µM, 0.85 ± 0.08 µM, 2.27 ± 0.17 µM and 2.56 ± 0.34 µM, respectively. The celecoxib was used as a standard compound with IC_50_ values 7.79 ± 0.84 µM for HeLa, 10.03 ± 0.84 µM for HepG2, 14.36 ± 0.96 µM for F10, and 15.64 ± 1.23 µM (Table 5) for A549, respectively. The analogue **43b** (Figure 7) exhibited the highest activity against the 293T cell line, with IC_50_ range 101.24 ± 2.27 µM and derivative **43c** (Figure 8) displaying strong activity against the L02 cell line with IC_50_ 104.57 ± 2.73 µM. Celecoxib was used as a reference drug with IC_50_ values 95.26 ± 2.28 µM against 293T, and 98.15 ± 2.39 µM against L02, respectively. The scaffold **43d** (Figure 7) showed the highest inhibitory activity with IC_50_ value 39.45 ± 1.33 µM against COX-1, while scaffold **43a** proved to be good inhibitor against COX-2 with IC_50_ = 0.09 ± 0.01 µM. Celecoxib was used as standard reference compound with IC_50_ value 43.37 ± 1.44 against COX-1 and 0.31 ± 0.12 (Table 6) against COX-2, respectively [71].

### 2.6. 3-Sulfamoyl Coumarins against Cancer-Related IX and XII Isoforms of hCAs

Dar’in et al. in 2021 synthesized series of seventeen 3-sulfonamide substituted coumarin scaffolds via the reaction of ethyl 2-sulfamoylacetate **45** and various substituted salicylaldehydes **44** in n-butanol at 110 °C for 2–6 h under basic conditions. The reaction mixture was stirred to obtain 3-sulfamoyl coumarines **46** in 30–84% yield (Figure 6) [72].

The 3-sulfonamide substituted coumarin derivatives screened for their in vitro metalloenzyme human carbonic anhydrase inhibitor, such as hCA I, II, IX, and XII. The 3-sulfonamide-substituted naphthalene coumaryl derivative **46a** (Figure 8) showed that the incorporation of extra aromatic rings may be detrimental for the anti-hCA activity of 3-sulfamoyl coumarins. The derivatives **46a** only showed micromolar KI values against all studies enzyme isoforms, as shown in Table 7, as compared with the reference drug AAZ (acetazolamide). The analogue **46a** also displayed good inhibition and retained its selectivity for the A431 cell line over non-tumorigenic normal human fibroblast cell line WI-26 AV4. The IC_50_ value of compound **46a** grows with an increasing incubation period (24 h to 72 h), from 9.73 ± 3.13 μM to 70.14 ± 26.06 μM, in comparison with standard reference drug Gefitinib (32.17 ± 3.44 μM, 16.02 ± 2.87 μM), as shown in Table 7. The promising compound **46a** was not the strongest CA IX/XII inhibitor, and apoptosis was induced by activating caspases in a dose-dependent manner. The DNA intercalation mechanism was followed by this compound **46a** for anti-proliferative effects, and so this compound has the potential to become a viable lead motif for the further development of next-generation anticancer agents [72].

### 2.7. Triazole-Bridged Coumarin Sulfonamides as CAs Inhibitors and Anticancer Agents

Kurt and coworkers in 2019 developed a synthetic approach to achieve the target coumarin sulfonamde motifs **53a**–**i** by consecutive reactions from various substrates for the evaluation of carbonic anhydrases (CA I, II, IX, and XII) inhibition and cytoxicities potential against human colorectal adenocarcinoma cell line HT-29 and healthy HEK293T embryonic kidney cells lines. In the first step, 7-hydroxy coumarin **47** was treated with 1,3-dibromopropane to furnish scaffold **48**, which was further treated with sodium azide to afford propylazide bearing coumarin derivative **49**. On the other hand, various aldehydes **50** hydroxy groups were combined with the propargyl group of propargyl bromide in DMF at rt to yield propargyl substituted aldehydes **51**, which were further treated with derivative **49** via a click reaction to obtain aldehyde-based triazole bridged coumarin derivative **52**, by using CuSO_4_·5H_2_O as a catalyst. The 4-(aminomethyl)benzenesulfonamide was reacted with the formyl group of compound **52** in alcoholic basic conditions, to achieve target triazole moiety containing coumarin sulfonamides **53**, as depicted in Figure 7 [73].

Kurt et al. in 2019 evaluated the triazole-bridged coumarin sulfonamide scaffolds against four physiologically significant isoforms CA I, II, IX, and XII of carbonic anhydrases by utilizing stopped flow carbon dioxide hydrase assay and anticancer activities against colorectal cancers HT-29 and healthy HEK293T embryonic kidney cells lines by MTT assay. CA IX is a prominent target for, in particular, colorectal cancers HT-29 cell line, which is overexpressed; this resulted in poor prognosis. The triazole-bridged coumrin sulfonamide **53a** (Figure 9) displayed significant and the highest hCA IX inhibition with the K_I_ of 45.5 nM in comparison with the standard reference drug AAZ (25.8 nM) as displayed in Table 8. The scaffold **53a** exhibited excellent anti-proliferative activity (IC_50_ 17.01 ± 1.35 μM) as compared with the reference drug doxorubicin (IC5.38 ± 1.40 μM), while this compound **53a** showed the least cytotoxicity (118.73 ± 1.19 μM) against healthy HEK293T embryonic kidney cells lines. On the basis of these findings, this novel compound **53a** cellular proliferates in human colon cancer cells by specifically targeting the CA IX and CA XII expression. The data indicated that the lead compound **53a** may be a promising drug candidate [73].

### 2.8. Substituted Coumarins Sulfonamide as Selective Human CA IX and XII Inhibitors

Abdelrahman and his colleagues in 2021 designed synthetic strategies to achieve different, substituted coumarin sulfonamide derivatives and screened for different biological activities and molecular dynamics. In the first synthetic approach, 3-acetylcoumarins **54** were condensed with 4-sulfamoylbenzoic acid hydrazide **55** and 4-hydrazino benzenesulfonamide **57** in boiling glacial acetic acid, to achieve 3-substituted coumarin sulfonamide derivatives **56a**,**b** and **58a**,**b**, as depicted in Figure 8 [74].

The coumarin sulfonamide hydrazones **56a**, **56b** and hydrazides **58a** and **58b** (Figure 10) displayed the potent hCA I inhibitory activity with inhibition constant K_I_ 98.8, 159.7, 77.6 and 92.5 nM in comparison with standard reference drug AAZ (K_I_ 250, 12, 25 and 5.7 nM), as described in Table 9. Sulfonamide moiety is essential in the hydrazone derivatives **56a**,**b** and hydrazide **58a**,**b** for the inhibition of both hCA I and II isoforms. The replacement of hydrazone moiety of scaffolds **58a**,**b** with hydrazide in scaffolds **56a**,**b** slightly enhanced inhibitory activity against hCA I and hCA II isoforms [74].

In the second synthetic approach, Abdelrahman and coworkers carried out the bromination of 3-acetylcoumarins **59a**,**b** with bromine in the presence of glacial acetic acid to furnish 3-(bromoacetyl)coumarins **60a**,**b**, which subsequently refluxed in ethanol with sodium benzenesulfinates **61a**,**b** to achieve phenylsulfonyl-based acetyl coumarin motifs **62a**,**b**. The coumarin arylsulfonehydrazones **63a**–**d** and hydrazide **64a**,**b** scaffolds were furnished in 74–82% yield via the treatment of compound **62a**,**b** with phenyl hydrazine and benzoic acid hydrazide in ethanolic glacial acetic acid solution and refluxed for 2 h, as shown in Figure 9 [74].

The coumarins, sulfolanamide **63a**,**b** and **64a**–**d** displayed excellent selectivity profiles, but these scaffolds have very little potential of inhibition; these derivatives barely inhibited hCA IX/XII carbonic anhydrases. These motifs inhibit the hCA II isoform up to 100 μM concentration.

In the final dual tail synthetic strategy (Figure 10), 4-sulfamoylbenzoic acid hydrazide, and 4-hydrazinobenzenesulfonamide were refluxed with 3-(2-(phenylsulfonyl) acetyl) coumarins **62a**,**b** in ethanol, to achieve coumarin sulfonamide arylsulfone hydrazide **65a**,**b** and coumarin sulfonamide arylsulfonehydrazones **66a**–**d** respectively, in the 74–78% yield as depicted in Figure 10 [74].

The incorporation of the arylsulfone moiety in the dual tail coumarin sulfonamide hydrazide **65a**,**b** and hydrazone **66a**–**d** (Figure 11)derivatives lowered the inhibition efficacy toward hCA I and hCA II isoforms in comparison to their counterpart derivatives **56** and **58**. The most selective dual tail coumarin sulfonamide derivatives were compounds **65a** and **66d**, as shown in Table 10. In the present study, hydrazides **58b** and **65a** proved the best and excellent CA IX inhibitors with low nanomolar potencies (9.8 and 18.6 nM, respectively) [75].

### 2.9. Coumarin Sulfonamide as RAF/MEK Inhibitors and Anticancer Agents

Aoki and coworkers synthesized compound **67** by the benzylation reaction of ethyl acetoacetate, which was further treated with 4-chlororesorcinol **68** by the Pechmann reaction to furnish substituted coumarin derivative **69** (Figure 11). On the other hand, the substituted coumarin aniline derivative **57** was obtained by the carbamoylation of the phenoic hydroxyl group with *N*,*N*-dimethyl carbamoyl chloride after the reduction of a nitro group was carried out by using SnCl_2_. The sulfamoyl chloride or sulfamoyl oxazolizinone was treated with scaffold **70**, which led to the formation of sulfamide **71**, as described in Figure 11 [75].

The series of newly reported coumarin containing sulfonamides was evaluated for their enzyme inhibition activities and tested for in vivo antitumor activity against HCT 116 (human colon cancer), MEK1, C-Raf, and HT-29 xenograft. The scaffold **71a** (Figure 12) showed the highest antitumor inhibition activities against MEK1, HCT116 and HT-29 with IC_50_ values 7 nM, 4 nM, and 1 nM, respectively. Moreover, **71b** (Figure 12) was the most potent against C-Raf, with an IC_50_ value of 5 nM. The derivatives **71c** and **72d** (Figure 12) displayed the highest antitumor activity with IC_50_ value 4 nM against C-Raf. The analogue **71e** (Figure 12) was the most active MEK1 inhibitor, with IC_50_ value 5 nM, while the derivative **71c** demonstrated the strongest anticancer activity with IC_50_ 23 nM (Table 11) against HCT1169, respectively [75].

## 3. Pyrazoline-Based Coumarin Sulfonamide Hybrids as Anticancer Agents

Amin and the team reported that synthetic methodology to furnish pyrazoline-based coumarin sulfonamide hybrids by refluxing acetic anhydride with 7-hydroxy substituted chromene **72** for 5 h, to produce substituted acetate chromenyl **73**, which was further heated with AlCl_3_ at 145 °C for 1 h to afford hydroxy acetyl chromen-2-one derivative **74**. By refluxing compound **74** and scaffold **75** in methyl iodide and dry acetone for 1 day to synthesize scaffold **76**, which reacted with aryl aldehyde and 10% NaOH in ethanol at rt for 1 day to yield substituted methoxy containing aryl acryloyl chromen **77**, which further proceeds in two steps. At the first step, derivatives **77** reacted with 4-substituted phenyl sulphonyl hydrazines **78** in absolute ethanol by refluxing at 8–12 h to afford pyrazoline-based coumarin sulfonamide hybrids **79** in 35–68% yield (Figure 12) [76].

The coumarin-pyrazoline containing phenyl sulfonyl moiety was screened for anticancer therapeutic efficacy against nine different tumor cell lines, including colon cancer, melanoma cancer, and breast cancer. The pyrazoline-based coumarin sulfonamide derivative **79a** (Figure 13) showed the best anticancer activities against tumor cell lines. The compound **79a** exhibited the highest antitumor activity against colon cancer (HCT-116), melanoma (LOX IMVI), and breast cancer (MCF7). Scaffold **79a** possessed the most powerful activity displaying the GI_50_ value 0.94 µM against HCT-116, 0.87 µM against LOX IMVI, and 0.49 µM (Table 12) against MCF7. The values were measured by different parameters for the three analogues GI_50_, TGI, and LC_50_ against every cell line. GI_50_ is the cytotoxicity parameter for the molar concentration of the scaffold that decreases 50% of the cell growth, and is observed as a growth-inhibitory level of effect; TGI (cytostatic activity) is the molar concentration of the derivative, leading to total growth inhibition; LC_50_ is the molar concentration of the compound that causes 50% net cell death. The SAR showed that the presence of electron-withdrawing chlorine atom on the fourth position of the phenyl sulfonyl structure and the phenyl group on the 5th position of the pyrazoline ring was responsible for the most potent antitumor activity of the scaffold **79a** [76].

### 3.1. Pyrazoline-Based Coumarin Sulfonamide Hybrids as Anticancer Agents

Zhang et al. in 2021 reported the synthesis of the coumarin sulfonamides derivatives **84** by the consecutive reactions of various substrates. In the first step, the substituted aromatic amines **80** were combined with methyl 2-chlorosulfonylacetate **81** rt by refluxing in TEA for 3 h to afford derivative *N*-phenyl sulfomyl acetate **82**. The scaffold sulfomyl acetae **82** further reacted with 4-(diethylamino) salicylaldehyde **83** in the presence of piperidine at 87 °C under reflux for 30 min to furnish substituted *N*-phenyl sulfomyl moiety containing diethyl amini coumarin motifs **84**, as depicted in Figure 13 [77].

The following coumarin-based sulfonamide derivatives were afforded and screened for their in vitro antitumor activity against MDA-MB-231 (human breast cancer cell line), HCT-116 (human colon cancer cell line), and KB (human oral epidermoid carcinoma cell line). The substituted *N*-phenyl sulfomyl moiety containing diethyl amini coumarin motifs **84a** (Figure 14) displayed the highest inhibitory anti-cancer efficacy against MDA-MB-231 cell line with IC_50_ value 9.33 ± 1.81 µM in comparison with the reference compounds 5-fluorouracil with IC_50_ 8.59 ± 0.52 µM (Table 13). The SAR investigated that the introduction of methoxy group on the benzene ring and diethylamino substituent at C-7 position and sulfonamide moiety at C-3 position of the coumarin enhanced antitumor activity. The compound **84a** had unique and versatile therapeutic efficacy, and properties against tumors, such as invasions, inducing apoptosis and the inhibition of cell migration. Scaffold **84a** upregulated the expression of caspase-3 and enhanced cancer cell apoptosis by increasing reactive oxygen species (ROS) level in MDA-MB-231 cells, as indicated by ROS assay and Western blotting analysis. The results of this study suggested that scaffold **84a** could be a promising lead anti-cancer agent which requires further exploration and research to become an antitumor drug [77].

### 3.2. Pyrazole Sulfonyl Coumarins Hybrids as Anticancer Agents and Anti-Migratory Activity

El-Sawy synthesized substituted sulfonyl coumarin derivatives via the treatment of sulfonyl chloride chromene **85** were refluxed with 2-acetyl-2-cyanoacetohydrazide using drops of triethylamine containing absolute ethanol to obtain substituted amino acetyl dihydropyrazol derivative **86** in 85% yield. Moreover, compound **85** and amino-based pyrazolinone reacted in the presence of dry dioxane in ethanol to afford sulfonic acid-substituted chromene-based pyrazolyl amide **87** in a 77% yield. In another approach, scaffold sulfonyl-based chromene malononitrile **88** in 76% yield was afforded by refluxing scaffold **85** with malononitrile. The derivative **88** further cyclized with hydrazine hydrate using triethylamine, along with dry ethanol, to produce pyrazole **89** in an 82% yield (Figure 14) [78].

The non-cytotoxic tested compounds were screened against HepG2 (hepatocellular carcinoma cells). The compounds significantly inhibited MMP-2 activity at *p* value ˂ 0.001 as a percentage of control. The results of this study demonstrated that the derivatives **86**, **87**, and **89** (Figure 15) are non-toxic angiogenesis inhibitors, and the compound **86** proved to be a promising anti-angiogenic agent. The coumarin sulfonamide derivative **89** displayed the highest MMP dependent (protease-dependent) anti-migratory activity (Table 14), as compared to all other tested compounds. The SAR showed that the presence of electron donating di-amino moiety on the pyrazole ring of scaffold **28** was responsible for the highest anti-migratory effect [78].

### 3.3. Coumarin Benzomidazole Sulfonamide Hybrids as Anticancer Agents

Holiyachi and coworkers in 2016 reported synthetic protocols to afford a series of substituted sulfonamide-based coumarin benzimidazole hybrids **93** with different substituents. The substituted 4-formylcoumarins **90** and ortho-phenylenediamine (OPD) **91** in DMF in the presence of *p*-Toluenesulphonic acid (*p*-TsOH) to achieve benzimidazole moiety contained substituted coumarin derivative **92** with an excellent yield 89–98%. The *N*-sulphonation of benzimidazole moiety of coumarin derivatives **92** was carried with *p*-toluenesulfonyl chloride (TsCl) in the presence of TEA to furnish the coumarin-benzimidazole sulfonamide hybrids **93** in a good to excellent yield—75–90% (Figure 15) [79].

The novel series of coumarin-benzimidazole sulfonamide hybrids (**93a**–**c**) was tested against different cancerous cell lines, such as HeLa (human cervix cancer) cell lines and human colon HT29 cancer cell lines. Among all these, the scaffolds **93a** (methyl coumarin) and **93b** (methoxy coumarin) exhibited the highest anticancer activity towards HeLa line with GI_50_ 36.2 and 35.3 respectively, while the analogues **93a** and **93d** (bromo coumarin) displayed the most powerful anticancer activity against HT when compared with the standard drug Adriamycin (ADR), respectively. The SAR highlighted that the presence of methyl, methoxy, and bromine substituent on the 6-position of coumarin ring enhanced the anticancer activity, as depicted in Figure 16 [79].

### 3.4. Coumarin-Proline Sulfonamide Motifs as Anticancer Agents

Durgapal and Soman in 2019 report the synthetic approach to achieve coumarin-proline sulfonamide derivatives **100**, and screened for their anticancer and antidiabetic therapeutic potential. In this synthetic strategy (Figure 16), the carbamate-protected 3-aminophenol **95** was afforded by the reaction of ethyl chloroformate with 3-aminophenol **94**. The 7-carboethoxy amino coumarin **96** was obtained by the Pechmann reaction of 3-aminophenol **94** with ethyl acetoacetate in 70% ethanolic H_2_SO_4._ The amino-substituted 4-methyl coumarin motifs **97** were afforded by the deprotection of derivative **96** in H_2_SO_4_/CH_3_COOH (1:1). The derivative **98** was achieved by the reaction of (L)-N-Boc-proline, first at 0–5 °C by stirring with ethyl chloroformate in tetrahydrofuran (THF) by adding derivative **97** solution and TEA dropwise, and then refluxed the mixture for 8 h in THF. The compound **98** contained the protection of the Boc group which was deprotected through trifluoroacetic acid (TFA) to obtain derivative 99, which was further treated with different substituted benzene sulfonyl chloride derivatives by using sodium hydrogen carbonate in CH_2_Cl_2_:H_2_O (1:1) at rt to furnish coumarin-proline sulfonamide derivatives **100** (Figure 16) [80].

The series of coumarin-proline sulfonamide derivatives were evaluated as anticancer against different cancer cell lines, such as the breast cancer cell line (MCF7), and a lung cancer cell line (A549), using the MTT assay and DPP-IV inhibition assay. The coumarin-proline sulfonamide scaffold **100a** (Figure 17) was the most potent against MCF-7 with an IC_50_ value of 1.07 mM, as compared to reference standard drug fluorouracil with an IC_50_ value of 45.04 mM (Table 15). The SAR showed that the activity of compound **100a** was significantly increased, due to the presence of methyl substituent at a parallel position of benzene ring to sulfonamide group [80].

### 3.5. Coumarin-6-Sulfonamide Derivatives as Anticancer Agents

Sabt and colleagues reported the synthesis of coumarin moiety containing sulfonamides and evaluated for apoptotic anti-proliferative activity. The chromene-based sulfonyl chloride **101** and 4-aminoacetophenone **102** reacted in pyridine in the presence of dichloromethane for 24 h at rt to yield acetyl phenyl-bearing chromene-based sulfonamide **103** that further reacted in two different steps. In this step, scaffold **103** refluxed with substituted phenyl hydrazine in the presence of absolute ethanol for 8 h to obtain coumarin-6-sulfonamide derivatives **104** in the 58–66% yield (Figure 17) [81].

The Sabt and coworkers synthesized coumarin containing substituted sulfonamide structural hybrids and were tested for apoptotic proliferation inhibition potential against the colon cancer cell line (Caco-2), hepatocellular carcinoma cell line (HepG2), and breast cancer cell line (MCF-7). The scaffold phenyl hydrazono containing coumarin-6-sulfonamide **104a** (Figure 18) showed the excellent and remarkable proliferation inhibition activity with IC_50_ value 8.53 ± 0.72 µM, in comparison with standard drug doxorubicin having IC_50_ = 4.10 ± 1.37 µM (Table 16) [81].

Sabt and colleagues prepared coumarin-6-sulfonamides (**107a**,**b** and **109a**,**b**) via the treatment of compound **103** with thiosemicarbazide using a catalytic amount of acetic acid in ethanol by refluxing for 6 h to furnish corresponding coumarin sulfonamide hydrazine carbothioamide **105** in a 34% yield. The intermediate **105** further refluxing with phenacyl bromide **106a** and coumarin-3-acetylbromide **106b** for 8 h using dioxane to achieve thiazoles moiety containing coumarin sulfonamides **107a** in an 80% and **107b** in a 63% yield. They further treated the intermediate **105** with bromoacetic acid **108a** and 2-bromopropanoic acid **108b** using anhydrous sodium acetate in acetic acid to afford the target thiazolidinone derivatives **109a**,**b** with a 40–64% yield (Figure 18) [81].

The thiazole moiety containing coumarin sulfonamide structural hybrids was evaluated against the MCF-7 cell line, HepG2 and Caco-2 cell lines for proliferation inhibition activities. The thiazole moiety containing coumarin sulfonamide analogue **107a** (Figure 19) displayed the best anti-proliferative activity against hepatocellular carcinoma cell line with a IC_50_ value 3.48 ± 0.28 µM (Table 17) in comparison with the standard reference drug doxorubicin of IC_50_ = 5.43 ± 0.24 µM among all other screened derivatives of this series. The methyl-substituted thiazole-based coumarin sulfonamide **109b** (Figure 19) was the most active derivative, which exhibited highly significant anticancer therapeutic potential against MCF7 cell line with IC_50_ value 10.62 ± 1.35 (Table 17), by comparing it with standard compound doxorubicin drug having an IC_50_ value 3.18 ± 0.32 µM [81].

### 3.6. Coumarin-Based Substituted Benzenesulfonamide Derivatives as Anticancer Agents

Debbabi and coworkers reported the series of substituted cyanoacetohydrazonoethyl methyl-based benzenesulfonamide derivatives (Figure 19) and evaluated for anticancer and antimicrobial therapeutic potential. The different coumarin-based benzenesulfonamides scaffolds **112** and **114** were achieved in dioxane under refluxing conditions via the reaction of cyanoacetyl containing hydrazonoethyl methyl-based benzenesulfonamide **110** with salicyldehyde **111** and 2-hydroxy naphthalene-1-carbaldehyde **113**, respectively (Figure 19) [82].

The series of all newly synthesized cyanoacetohydrazonoethyl-*N*-ethyl-*N*-methyl benzenesulfonamide derivatives were evaluated for anti-cancer therapeutic potential against the MCF cell line. The coumarin moiety containing a benzenesulfonamide **112** scaffold (Figure 20) displayed significant and remarkably high anticancer activity against the MCF-7 cell line, with the IC50 value 1.08 µg/mL (Table 18) among all synthesized scaffolds. The MTX (methotrexate) was used as a standard compound with an IC_50_ value of 12.3 µg/mL (Table 18) against MCF-7 [82].

## 4. Conclusions

The conducted review of coumarin sulfonamide derivatives and their structural hybrids with various substitution patterns displayed the anticancer and carbonic anhydrases’ different isoforms inhibition potential against a wide variety of cancer cell lines. Cancer is considered a leading cause of high mortality and morbidity. A great number of different anticancer and carbonic anhydrase inhibitors were developed, but due to rapidly evolving drug resistance, reduced the efficacy of curing various types of cancers. Most of the coumarin sulfonamides and benzene sulfonamides containing coumarin moiety derivatives cited in this review are the most active or nearly equal in the efficacy and therapeutic potential, in comparison with marketed anticancer drug and enzyme inhibitors. The SAR analysis provides information to develop novel coumarin sulfonamide derivatives and structural hybrids with improved efficacy, better therapeutic potential, and lower side effects.

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
