# Peer review of "An Update on Synthesis of Coumarin Sulfonamides as Enzyme Inhibitors and Anticancer Agents"

_molecules, 2022, doi:10.3390/molecules27051604_

Round 1

Reviewer 1 Report

  • On Pg-3 line 107, Derivative 7  reacts with various sulphonamides should be derivative 8 as shown in its scheme.
  • Literature review for Coumarin based sulfonamides is not complete. It has covered only upto 2017-18. Papers published on coumarin based sulphonamides after 2018 are not covered, which should be included. In 2019  coumarin based proline sulfonamides as anticancer , andantidiabetic agents,  work is published ,which should be covered.
  • On Page 10, Scheme-5 , Compound 30 ,is it simple 7-hydroxy coumarin or -hydroxy 4-methyl coumarin? Please check.
  • On Page 11 , line 268, and 270, for 41 and 43,  is it condensation reaction? It should be substitution reaction.
  • On Page 12 ,Scheme 6 reaction conditions  are not mentioned, which should be included.

Author Response

Dear Reviewer,

Thank you for your comments and valuable suggestions.

Yours sincerely, 

Mariusz Mojzych

Reviewer 2 Report

Its unique features and the vast possibilities of functionalization, makes the coumarin a privileged scaffold in the medicinal chemistry. Accordingly, this versatility has translated into a vast range of biological activities, such as antiviral, anticholinesterase antioxidant, anticoagulant, and anticancer activities.

The current review entitled "An update on synthesis of coumarin sulfonamides as enzyme inhibitors and anticancer agents" by Rubab et al, is dedicated to present an overview about the relevant studies described the diverse anticancer activities for the coumarin sulfonamide-based small molecules. I can’t recommend this review to be published in Molecules in its current form. Authors should address the following points.

- Authors stated in the last paragraph in the introduction section “This review article provides comprehensive overall in current progress of coumarin sulfonamides in the medicinal chemistry as carbonic anhydrase inhibitor, anticancer and other medicinal agents”. Surprisingly, we can find that this review covers only NINE studies (from Ref 67 to Ref 75).The remaining references were cited in the introduction section. For sure 9 studies are not enough to prepare a comprehensive work and to shed the light in the topic under investigation. Authors should cover more studies.

- The organization and subtitles of the manuscript are not clear.

- Regarding the title: 3.0. Cyanoacetohydrazonoethyl methyl based benzenesulfonamide derivatives as anticancer agents. I see it has no relation to the current review. Only two structures from this study incorporated the coumarin moiety.

- There is no any cited reference in 2021!!

- Also, there are only 3 references cited from 2020!!

- A good review not only conveys what authors have read, but also should take the next step and comes up with an angle that extends what they have read. What does it mean to take the next step? In other words, authors should try to analyze and evaluate, not report and list. It is apparent in the text that authors mentioned a lot of unnecessary details to extend the size of the review.

- The structure for the reported molecules in this review should be revised thoroughly, there are some incorrect structure, such as compound 20.

- The orientation of the coumarin moiety should be the same for all compounds in all the presented figures. This issue could be explained by figure 4. Also, we can find that compounds 9 have different orientations in Scheme 1 and Figure 3!!

- Some structures need “clean up structure” in the ChemDraw.

- In most Tables, the column “Serial Number” has no role and should be removed. Like Tables 1 and 3.

- The reagent and conditions for many schemes lack many items, such as the reaction temperature and duration. Please revise thoroughly.

- The language of the manuscript needs extensive editing and revision.

Author Response

(The authors gave the same response as above.)

Reviewer 3 Report

The ms of Irfan/ Mojzych et al gives a nice overview on coumarin sulfonamides including synthetic and bioactive aspects. The ms may be suitable for Molecules after the following minor revisions.

  • 2/3, Scheme 1 (and text): there is no need to give the yields to decimals. E.g. 90.6% should be 91%. This should be corrected throughout.
  • RT may be 18-35 °C. Pls precise (in and) above Scheme 1, as well as in Schemes 4, 5, 7, 8 and 9.
  • The phrases „acid-based, coumarin-containing, thiazole-based, pyrano-substituted, methoxy-containing, pyrazoline-based, pyrazole-based, amino-based, methyl-based” should all be written using „hyphen”.
  • 3, l.106: „DMF (dimethylformamide)” should be in a reversed oder.
  • instead of „benzene sulfonyl” pls write „benzenesulfonyl” throughout.
  • 6, l. 164: instead of „on benzene ring” write „in benzene ring”
  • 9/bottom and p.11, l.269: „produce” is not a good word here. Vegetables may be produced.
  • Scheme 6: pls provide conditions
  • Scheme 12: instead of „S” pls write „S8
  • In a part of the schemes, the exact conditions are missing. Pls insert in Schemes 1,3,11 and 12.
  • Conclusion should be in plural.
  • Instead of „Scheme -1”, „Figure-1”, etc. pls use the formula without „hyphen”. This should be corrected throughout.
  • The references should be checked, pls see the duplums:

„Curr. Pharm. Des”         „Current Pharmaceutical Design”

„ Nat. Prod. Rep.”                        „Natural products reports”, etc.

Author Response

(The authors gave the same response as above.)

Round 2

Reviewer 2 Report

The manuscript has been improved.